# Performance Enhancement of Opened Resonance Photoacoustic Cells Based on Three Dimensional Topology Optimization

Zihao Tang [1], Wenjun Ni [1], Zehao Li [1], Jin Hou [1], Shaoping Chen [1], Perry Ping Shum [2] and Chunyong Yang [1,*]

1 Hubei Key Laboratory of Intelligent Wireless Communications, Hubei Engineering Research Center of Intelligent Internet of Things Technology, College of Electronics and Information Engineering, South-Central University for Nationalities, Wuhan 430074, China; 201821111386@mail.scuec.edu.cn (Z.T.); wjni@mail.scuec.edu.cn (W.N.); 201721113048@mail.scuec.edu.cn (Z.L.); houjin@mail.scuec.edu.cn (J.H.); spchen@scuec.edu.cn (S.C.)

2 Department of Electrical and Electronic Engineering, College of Engineering, Southern University of Science and Technology, Shenzhen 518055, China; shenp@sustech.edu.cn

* Correspondence: cyyang@mail.scuec.edu.cn

**Abstract:** Photoacoustic (PA) spectroscopy techniques enable the detection of trace substances. However, lower threshold detection requirements are increasingly common in practical applications. Thus, we propose a systematic geometry topology optimization approach on a PA cell to enhance the intensity of its detection signal. The model of topology optimization and pressure acoustics in the finite element method was exploited to construct a PA cell and then acquire the optimal structure. In the assessment, a thermo-acoustic model was constructed to properly simulate the frequency response over the range of 0–70 kHz and the temperature field distribution. The simulation results revealed that the acoustic gain of the optimized cell was 2.7 and 1.3 times higher than conventional cells near 25 and 52 kHz, respectively. Moreover, the optimized PA cell achieved a lower threshold detection over a wide frequency range. Ultimately, this study paves a new way for designing and optimizing the geometry of multifarious high-sensitivity PA sensors.

**Keywords:** optical sensors; photoacoustic cell; topology optimization; simulation

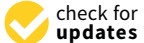



## 1. Introduction

Photoacoustic spectroscopy (PAS) is a typical analytical tool employed to probe the information of chemical composites by measuring the optical absorption characteristics of samples [1–5]. This technology has the advantages of high sensitivity, high selectivity, rapid response, and high precision, which are essential for online monitoring. Thus, its application and scientific research prospects are widely considered in the fields of agricultural ecology [6], environmental monitoring [7], chemical environmental protection [8], biomedical research [9], and others. Particularly in biological sample monitoring, mid-infrared PAS is a promising technique for noninvasively monitoring glucose by probing interstitial fluid through the skin [10–12]. The major technical challenge of the aforementioned PAS technology is the detection of weak acoustic waves generated from the thermal de-excitation of molecules upon the absorption of modulated electromagnetic radiation.

In PAS detection, the amplitudes of the PA signals are proportional to the incident light intensity, microphone sensitivity, and cell constant [13]. In fact, a PA cell can serve as a module to promote the light–matter interaction. More specifically, the molecules of the measured substance absorb light radiation and then undergo photo–thermal–acoustic energy conversion [14]. It is known that resonant and non-resonant patterns can be clarified according to the working mode of the cell [15]. Resonant cells markedly increase the intensity of PA signals. The intrinsic frequency of the resonant cell should match the modulated frequency of the incident light [15]. To date, a considerable amount of research works have focused on the shape modification of photoacoustic cells. In 2014,

M. A. Pleitez et al. [16] exploited a closed T-type cell for noninvasively monitoring glucose. This cell has two perpendicularly connected, cylindrical resonance and absorption cavities. Specially, the absorption cavity is sealed by an optical window at one end and opened to samples at the other end. In the same year, they created a windowless resonant cell that is opened at both ends (two-end-opened) [17]. Compared with the aforementioned closed cell, this opened cell can reduce the influence of temperature fluctuations, air pressure variations, and air humidity buildup, which allows the sensitivity and long-term stability of the cell to be accordingly increased. Since then, the opened type cell has attracted intensive attention in the context of the biological sample detection [18–21]. For example, J. Sim et al. [18,19] proposed a dominant resonance mode to match a microphone and opened cell. This method was able to effectively improve the signal-to-noise-ratio (SNR) and reduce the detection limit. In addition, the optimization of the size and shape of traditional H- and T-type cells has resulted in a wide use of PA cells in trace gas detection [22–26]. For example, X. Yin et al. designed a differential PA cell with a large linear dynamic range on sub-ppb level nitrogen dioxide detection [24]. In these reports, researchers have concentrated on the optimization of different shapes of PA cells and the acquisition of a lower detection limit.

However, the aforementioned approaches are hampered by the lack of a systematic method to obtain a good conceptual optimization design. To the best of our knowledge, topology optimization has gradually become a strong simulation alternative due to its unlimited freedom and marked solving capability in conceptual design. The purpose of topology optimization is to gain an optimal distribution of material resources across a design domain. For example, Haouari R. et al. [27] employed 3D topology optimization method to get a "potato-like" PA cell for trace gas detection, and the acoustic gain of the optimized cell was approximately twice than the conventional one. As a consequence, 3D topology optimization method can enhance the Q value and the pressure wave amplitude of the PA cells. Moreover, it is worth noting that the rapid development of 3D printing technology [28] will shine new light on the multidisciplinary practical applications of topology optimization.

In this work, taking noninvasively glucose monitoring as an example, a topology optimization method is proposed to strengthen the entire performance of a conventional T-type, opened PA cell, which usually provides poor detection sensitivity due to signal leakage at the resonator opening. A reverse engineering modeling was employed to smooth the surface of the opened container, which is referred to a "cardiac-type" resonance PA cell. Based on thermoviscous acoustics theory, we evaluated the cell by analyzing the frequency response and temperature field distribution over the frequency range of 0–70 kHz and temperature range of 273.15–323.15 K, respectively. The simulation results illustrate that the "cardiac-type" cell performed significantly better than the conventional one with respect to amplification gain and Q value. This topologically optimized PA cell is capable of achieving a high detection performance, which will pave a new way for highly sensitive nondestructive detection.

## 2. Principle of Topology Optimization and PA Cell Model

In general, topology optimization is used to search for the optimal distribution of specified materials in the design domain. To be specific, the material properties of each discrete spatial point are regarded as design variables that need to be optimized in the structure. In terms of a standard finite element model, the density approach introduces an element design variable that controls the properties of the specific material using the gradient-based optimization algorithm. Here, the design variable $\xi$, which represents the material relative density, is expressed as:

$$0 \leq \xi(x) \leq 1 \quad \forall x \in \Omega_d \tag{1}$$

where $\xi$ is normally relevant with position $x$ in the design domain $\Omega_d$, "0" indicates a void like air, and "1" is a solid material with a high specific acoustic impedance. Other intermediate gray values of the $\xi(x)$ variable represent mixtures of the two materials, which are

meaningless in physics and difficult to manufacture. Hence, the actually optimized solution is to determine the optimal "0–1" distribution of the two prescribed materials.

As defined in Equation (1), since the design variable $\xi$ could theoretically be any value between 0 and 1, optimization results may be dominated by numerical instabilities such as checkerboard patterns and mesh dependency. Thus, we designed a low-pass density filter that defines the physical density of an element $\bar{\xi}$ as a weighted average of the design variables $\xi$ in the radius $R$ region [29]. This density filter enables the raw scalar function $\xi$ to generate a smooth and well-behaved function of variable $\bar{\xi}$, and it can accelerate the convergence to 0–1 solutions throughout the optimization process. It can be implemented via a Helmholtz-type partial differential equation:

$$- R^2 \nabla^2 \bar{\xi} + \bar{\xi} = \xi \tag{2}$$

where $R$ represents the filtering radius. Generally, its value is adjusted to fit with the finite mesh element length from 1 to 1.5 times. However, the density filter in Equation (2) has the problem of generating gray (intermediate) transition material between solid and void (air) regions. To produce binary (black and white exclusively) designs, a volume preserving projection based on a smoothed Heaviside function can be used [30]. Particularly, the density filter variable $\bar{\xi}$ was projected onto a new design variable axis to produce as many gray regions as possible. The Heaviside function is expressed as:

$$\zeta = \zeta\big(\bar{\xi}(\xi)\big) = \frac{tanh(\beta\eta) + tanh(\beta(\bar{\xi} - \eta))}{tanh(\beta\eta) + tanh(\beta(1 - \eta))} \tag{3}$$

where $\beta$ is the projection slope that controls the projection level of the filtered design variable $\bar{\xi}$ into a 0/1 space and $\eta$ is the projection point that determines the threshold. As indicated in Figure 1b, when the value of the projection variable $\zeta$ is higher or lower than $\eta(0.5)$ (as illustrated by the dotted line), it immediately trends to 1 or 0 as $\beta$ increases, respectively. The authors of [31] discussed in detail how $\beta$ and $\eta$ values influence 0/1 projection results.

The gas in a PA cell is assumed to be an isotropic, uniform, lossless and initially stationary fluid medium. Thus, the weak acoustic wave propagation in a PA cell is governed by the Helmholtz equation:

$$\nabla\left(\frac{\nabla p}{\rho(\zeta)}\right) + \frac{\omega^2}{\rho(\zeta)c(\zeta)^2} p = 0 \tag{4}$$

where $\rho$, $p$, $c$, and $\omega$ represent the material density, complex pressure field, sound velocity, and free-space angular frequency, respectively. Here, $\rho$ and $c$ are the variables associated with the function $\zeta$. Generally, the Solid Isotropic Material with Penalization (SIMP) scheme is used to characterize material properties, which can be expressed as follows [32]:

$$\begin{cases} \rho(\zeta) = \rho_{air} + \zeta^q(\rho_{solid} - \rho_{air}) \\ c(\zeta) = c_{air} + \zeta^q(c_{solid} - c_{air}) \end{cases} \tag{5}$$

where $q$ is a penalty factor that can promote the intermediate values of the projection variable $\zeta$ approaching "0" or "1" as soon as possible. SIMP interpolation can eliminate intermediate material properties in a final design. When $\zeta$ is "0", the associated element corresponds to air and then the incident acoustic wave propagates the element to the other side. In contrast, a solid element whose density and sound velocity are extremely higher than the air is considered when $\zeta$ is "1". This solid element can almost completely reflect the incident acoustic wave. When $\zeta$ is other values between "0" and "1", the penalty factor $q$ effectively penalizes the specific element so that they gather around the boundary value. The effect of the binary approximation of material properties (density and sound velocity) as $q$ evolves is clearly shown in Figure 1c. It should be emphasized that the penalty effect is obvious when $q$ is selected to be a higher value; however, the fact that the optimization

process rapidly converges may result in worse multiple local optima. To acquire the highest sound pressure intensity at the microphone position, the objective optimization function takes the following form:

$$max \frac{1}{\int_{\Omega_d} dx} \int_{\Omega_d} |p(\zeta(\xi))|^2 dx \qquad (6)$$

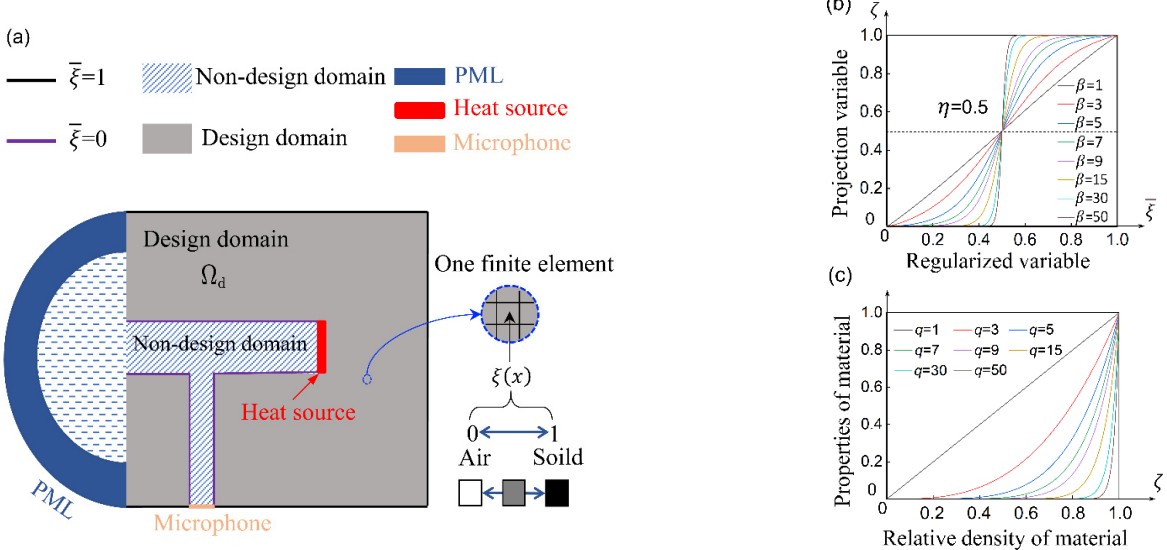

**Figure 1.** (**a**) Schematic of finite element topology modeling for a T-type opened PA cell. (**b**) Effect of modified Heaviside function with different values of parameter $\beta$ at $\eta$ = 0.5. (**c**) Penalization intensity corresponding to different values of penalty factor $q$.

In addition to the objective function, volume constraints are typically imposed on the design domain to modify the function. Specifically, the constraint reflects some inherent size limitations for optimization to avoid trivial solutions and save cost. The volume constrains over the design domain $\Omega_d$ are determined as follows:

$$\frac{1}{\int_{\Omega_d} dx} \int_{\Omega_d} \zeta(x) dx \leq k \qquad (7)$$

where $k$ is a prescribed volume fraction of allowable material with values between 0 and 1, where $k$ = 1 corresponds to no limit.

The calculations are repeatedly conducted during the iterative optimization process, and the computational efficiency is proportional to the cube of the number of design variables. Considering the symmetry of a PA cell and expensive computing resources, we designed a longitudinally symmetric 3D model that led to a half computational load. Figure 1a illustrates the 2D schematic model of the T-type opened cell configuration. The design variable $\zeta(x)$ that ranges from 0 to 1 was assigned to each finite element within the domain of interest. The gray and white areas with stripes are the design and non-design domains, respectively. Here, the testing sample was simplified to be a tiny cylinder-shaped heat source to simulate the thermal de-excitation of the specified molecules in the sample after laser excitation. The testing samplewas located at the end of the absorption cavity and was confined within the opening cross section. The miniature microphone was placed at the bottom of the cell to detect acoustic wave signals. The cell ventilation opening was modeled by using a hemisphere to represent the free space. The perfectly matched layer (PML) covered the hemisphere, which exponentially attenuated the reflection of the radiated acoustic waves. The radius of the hemisphere was 3 times the radius of the absorption cavity. The thickness of the PML was one third of the radius of the hemisphere,

which could achieve an excellent absorption effect. Moreover, the boundary of the design domain needed to be provided appropriate constraint conditions to better promote the converge of the model. In Figure 1a, the black and purple lines represent the boundary conditions of the function $\bar{\bar{\zeta}}$ and respectively correspond to $\bar{\bar{\zeta}} = 1$ (solid) and $\bar{\bar{\zeta}} = 0$ (air). The material of the design domain was selected to be polymethyl methacrylate (PMMA), which is easy to use in 3D printing. The density and sound velocity of the PMMA were 1180 kg/m$^3$ and 2500 m/s, respectively.

The target frequency was 25 kHz. This frequency was chosen because the ultrasound range is usually used for noninvasively monitoring glucose [17] and reduces most sound pollution. Moreover, the lower the frequency, the higher the PA signal [33]. We studied the obtained shapes depending on the evolution of several parameters. We set the evolution of the filtering radius $R$ in Equation (2) to different lengths of the finite mesh $h$, as well as the projection slope $\beta$ in Equation (3) and the penalty parameter $q$ in Equation (5). The density variable $\zeta$, whose initial value is usually 0.5, fulfilled the first iteration of the optimization procedure according to the maximization objective function in Equation (6) and constraint in Equation (7). We used an absolute tolerance of 0.001 on the maximum change of the design variables to terminate the optimization loop. The maximum number of iterations was set as 150. The design sensitivity for the objective optimization function was derived with the adjoint method [34] and was implemented via highly efficient vectorization techniques. Finally, we utilized the method of moving asymptotes (MMA) [35], which is a gradient-based algorithm that uses information from the previous iteration steps and gradient, as the solver.

## 3. Topology Optimization Result and Discussion

We changed parameters $R$ from 0.5 to 2 in increments of 0.1, and we changed $q$ and $\beta$ from 1 to 20 in increments of 1. We selected the optimal results using the spatial distribution of the projection function $\zeta$ as an indicator (as shown in Figure 2). The left side of each column in Figure 2 represents a 2D distribution, and the white and dark areas indicate air and PMMA, respectively. The right side of each column is a 3D diagram. We observed that the filtering radius $R$ had a great impact on the final optimized shape, which indicated that the optimization solution depended on the division of the mesh. When the $R$ value was small enough (0.5 $h$), the singularity of the optimization results appeared in the iteration process. In contrast, the optimized shape approached the original design, which may be effectless in the optimization process. By comparing simulation results, the optimal result may locate in the column of $R = 1$ $h$ or $R = 2$ $h$. It is difficult to obtain a mesh-independent design at high frequencies because the optimization is sensitive to the discretization, filtering radius, starting guess, and local maximum. In the optimal $R$ interval, the optimized designs were found to have close resemblance in overall topology but differed in structural details. In other words, well-defined structures seemed to automatically appear and there was no particular need for penalization and the projection of intermediate design variables. For the optimization problem of a strict reflection wave, this situation may occur because intermediate design variables reduce the contrast among the properties of materials, thus leading to a reduced reflection [30].

Furthermore, the intermediate values could be effectively penalized when the penalty factor $q$ was between 3 and 5. When $q$ was higher than 5, the surface of the optimized shape presented many small holes, thus highlighting an excessive penalty effect. By trading-off $q$, $\beta$, and $R$, the desirable structure in the red box in Figure 2 was selected as the final optimization shape, and their values are corresponding to 3, 5, and 1 h, respectively. Then, the final shape was guided into reverse engineering modeling software to smooth the surface. Finally, the processed shape was imported into finite element simulation software as a 3D design blueprint to analyze the frequency response and temperature field distribution. Here, the optimized structure is referred to as a "cardiac-type" PA cell.

According to the well-known Rosencwaig–Gersho model [33], PA signal amplitude ($A_{PA}$) shows the following dependence:

$$A_{PA} = P\frac{(\gamma - 1)Q}{Af\pi^2}\alpha M \tag{8}$$

where $P$ represents the output light power of the radiation source (given in W), $\gamma$ denotes the ratio of the specific heat capacity of the carrier gas at constant pressure to constant volume, Q is the quality factor of the resonator (which is discussed in detail below), $A$ is the cross-section area of the resonator (given in mm$^2$), $f$ is the modulation frequency of light source (given in Hz) that is equal to the selected resonance frequency of a PA cell, $\alpha$ is the optical absorption coefficient of the measured species (given in m$^{-1}$), and $M$ is the microphone sensitivity (given in $\mu$V/Pa).

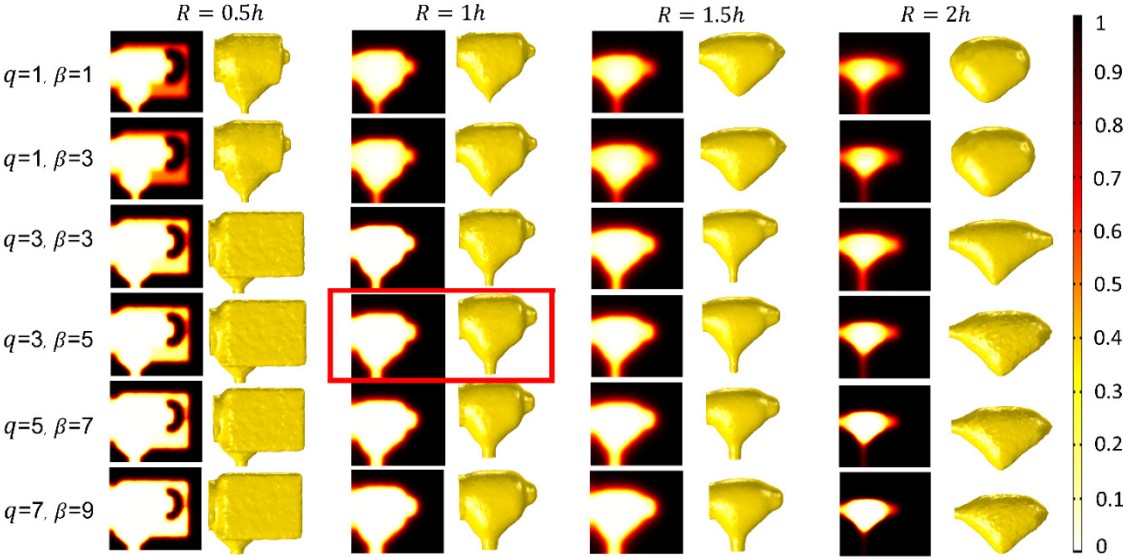

**Figure 2.** 2D and 3D reconstructed view list of the optimized shape with different $q$, $\beta$, and $R$ parameters.

The Q factor is defined as the ratio between the resonance frequency and the width of the resonance peak at half its resonance amplitude (FWHM). It is estimated as follows:

$$Q = \frac{f_0}{\Delta f} \tag{9}$$

where $f_0$ denotes the resonance frequency and $\Delta f$ is the full width at half maximum of the acoustic resonance [36]. As Q increases, the losses of the system decrease.

Equation (8) implies that a small-sized PA cell and a low modulation frequency are more desirable for obtaining strong PA signals. Additionally, in order to compare the performance of the two cells, we had to fix their light power and volume. In this work, we set the volume of the "cardiac-type" cell, conventional cell, and their sample cavities as 219.9, 219.32, and 2.2 mm$^3$, respectively. In order to avoid the influence of external environment, all the parameters of the two PA cells are set as the same except for the structure size. The details of the structural dimensions for both cells are shown in Figure 3.

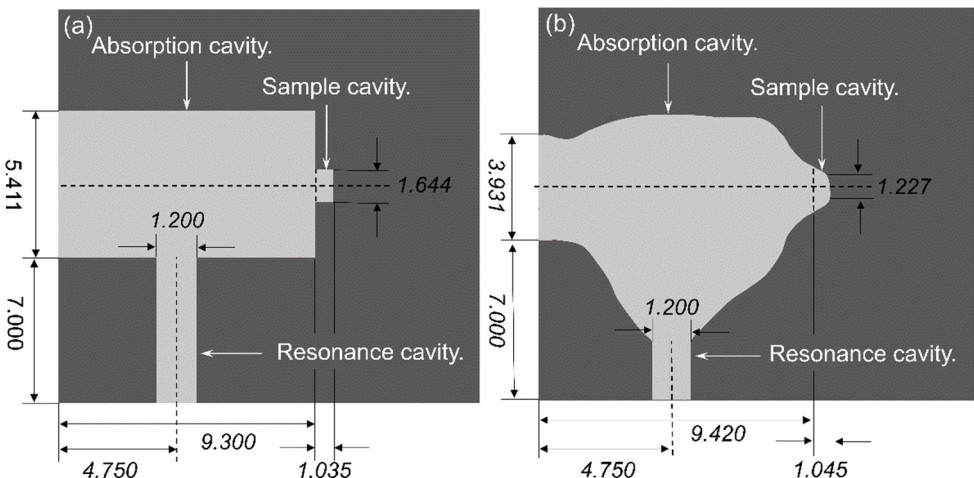

**Figure 3.** The schematic of two cells cross-section (shown as light grey areas): (**a**) conventional cell and (**b**) "cardiac-type" cell. All numbers have the unit of mm.

Generally, three main sources dominate the loss of an opened PA cell: surface loss, volumetric loss, and escaping loss. In terms of small resonators, surface loss is the most important factor that needs to be considered in a simulation. Thus, the viscous and thermal dissipation on the smooth internal surfaces must be accurately calculated. For this reason, a thermoviscous acoustics module was utilized to solve the full linearized Navier–Stokes, continuity, and energy equations.

$$\frac{\partial \rho}{\partial t} + \rho(\nabla u) = 0 \tag{10}$$

$$\rho\frac{\partial u}{\partial t} = \nabla \cdot \left[ -pI + \mu(\nabla u + (\nabla u)^T - \left(\frac{2}{3}\mu - \mu_b\right)(\nabla \cdot u)I \right] \tag{11}$$

$$\rho C_p \frac{\partial T}{\partial t} - \alpha_p T \frac{\partial p}{\partial t} = -\nabla \cdot (k\nabla T) + Q \tag{12}$$

where $\rho$, $u$, $p$, $\alpha_p$, $\mu$, $\mu_b$, $T$, and $C_p$ are density, acoustic velocity field, pressure, thermal expansion coefficient, dynamic viscosity, bulk viscosity, temperature, and heat capacity at a constant pressure, respectively. The values of the input parameters in the model are presented in Table 1.

**Table 1.** The values of the input parameters in the model.

| Parameters | Value | Unit |
|:---:|:---:|:---:|
| Density $\rho$ | 1.2046 | kg/m$^3$ |
| Pressure $p$ | $1.01325 \times 10^5$ | Pa |
| Temperature $T$ | 293.15 | K |
| Bulk viscosity $\mu_b$ | $1.086 \times 10^{-5}$ | Pa $\cdot$ s |
| Dynamic viscosity $\mu$ | $1.81 \times 10^{-5}$ | Pa $\cdot$ s |
| Acoustic velocity field $u$ | 343.37 | m/s |
| Coefficient of thermal expansion $\alpha_p$ | $3.67 \times 10^{-3}$ | 1/K |
| Heat capacity at constant pressure $C_p$ | 1005.42 | J/(m$^3 \cdot$ K) |

The thickness of viscous boundary layer $\delta_v$ and the thermal boundary layer $\delta_{th}$ are respectively defined as:

$$\delta_v = \sqrt{\frac{\mu}{\pi f \rho}} \tag{13}$$

$$\delta_{th} = \sqrt{\frac{k}{\pi f \rho C_p}} \tag{14}$$

where $k$ is the wavenumber. The boundary layers should be resolved by finite element mesh in a simulation. Here, a hybrid generation technique was adopted to control the mesh size and boundary layer. In the conventional cell, the resonance cavity, PML, and parts of the absorption cavity needed to exploit the swept meshes to save the calculation load and accurate input, as shown in Figure 4. Additionally, other areas of the cell were calibrated by a free tetrahedral mesh. In contrast, the "cardiac-type" cell only needed free tetrahedral mesh to divide its irregular shape. It should be noted that the boundary layers were generated along the inner walls in both cells to accurately map the loss effects of the thermal and viscous surface.

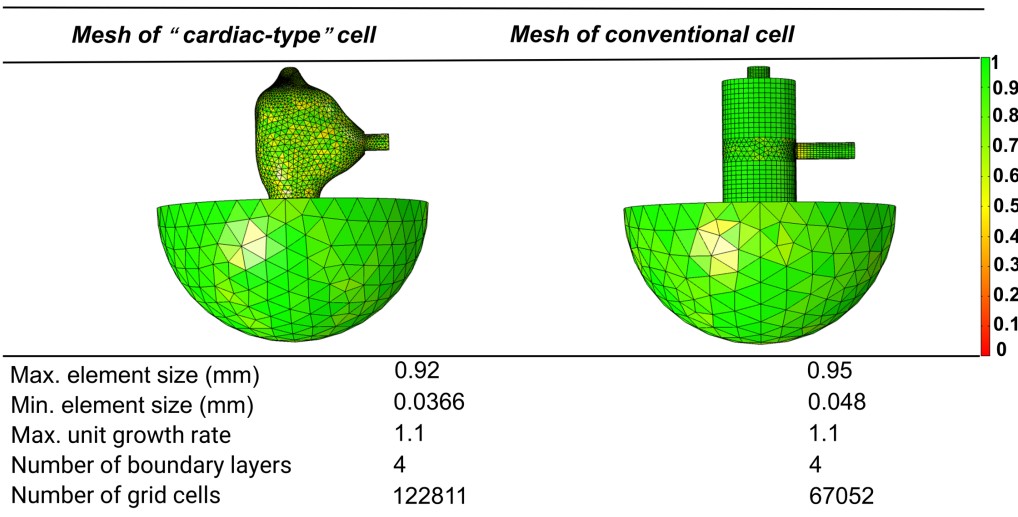

| | Mesh of "cardiac-type" cell | Mesh of conventional cell |
|---|---|---|
| Max. element size (mm) | 0.92 | 0.95 |
| Min. element size (mm) | 0.0366 | 0.048 |
| Max. unit growth rate | 1.1 | 1.1 |
| Number of boundary layers | 4 | 4 |
| Number of grid cells | 122811 | 67052 |

**Figure 4.** The properties of the two cell meshes.

In the simulation, the physical properties of the inner wall needed to keep no-slip and isothermal conditions. The power density of the heat source was set to $10^6 \ W/m^3$, and the swept-frequency was set from 0 to 70 kHz with a step of 50 Hz. The frequency response curves shown in Figure 5 illustrate that the "cardiac-type" cell had higher amplification capability at the resonance frequency and more resonance peaks near the receiving point. In addition, the cell exhibited narrower resonance widths, which indicates that lower loss was induced by the inner wall. The highest peak reflected that the optimization frequency located at 25 kHz for the "cardiac-type" cell. The sound pressure amplitude of the "cardiac-type" cell at 24.9 and 51.8 kHz is approximately 2.7 and 1.8 times higher than that of the conventional cell at 17.5 and 53.05 kHz, respectively. The Q value was accordingly higher by 1.9 and 4.9 times, respectively. Table 2 compares the partial Q factor obtained in this study and previous work. Compared with the reference PA cell, the performance enhancement of the optimized one is mainly attributed to the cardiac-type structure. We believe that the sound pressure enhancement was caused by the acoustic signal receiving a better impedance match with the receiving point. Meanwhile, such geometry was able to enable multiple reflected sound waves at the microphone position. Thus, the total measured sound pressure was greatly improved.

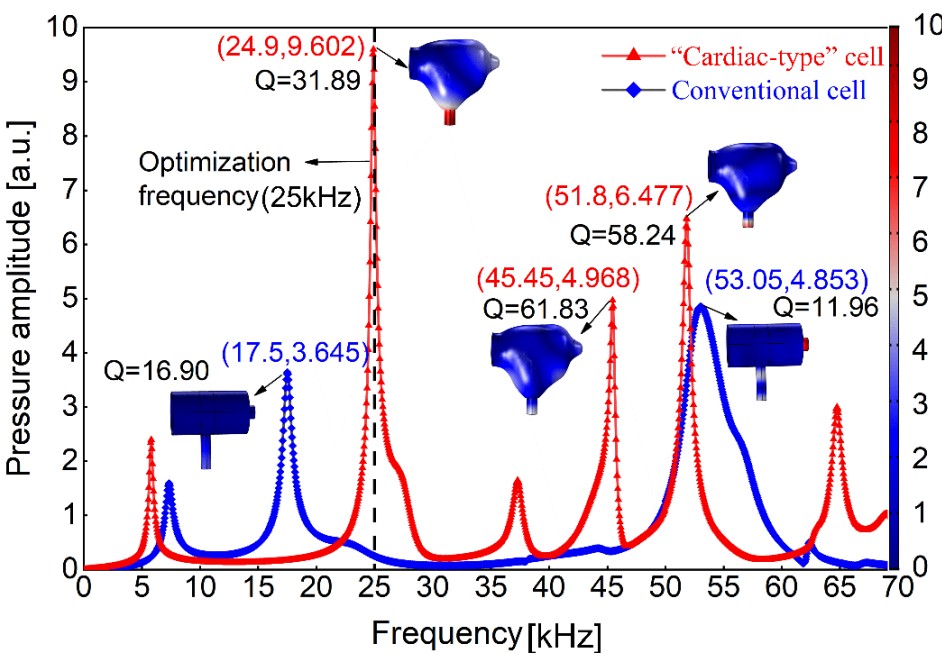

**Figure 5.** Frequency response of "cardiac-type" and conventional cells.

**Table 2.** A comparison between the Q factor values obtained in this study and previous work.

| Design on Opened PA Cell | Q#1 | Q#2 | Simulation/Experiment |
|---|---|---|---|
| This paper—conventional one | 16.9 (at 17.5 kHz) | 12.0 (at 53.1 kHz) | Simulation |
| This paper—optimized one | 61.8 (at 45.5 kHz) | 58.2 (at 51.8 kHz) | Simulation |
| El-Busaidy, S. et al. [20] | 51.0 (at 28 kHz) | 30.0 (at 49.8 kHz) | Simulation |
| Sim, J. et al. [19] | 6.1 (at 16.0 kHz) | 11.8 (at 47.0 kHz) | Experiment |
| Pleitez, M. et al. [17] | 45.0 (at 51.7 kHz) | 32.0 (at 53.8 kHz) | Experiment |

Characteristic frequency is a vital parameter for photoacoustic cells, and temperature may cause frequency shifts. If a PA cell is a thermal insulator, temperature variation can be ignored. However, the selected material based on the 3D printing technique for PA cells is relatively sensitive to temperature variation. Therefore, we needed to investigate whether the temperature induced major parameter variations in the PA cell, especially for the characteristic frequency. As previously mentioned, characteristic frequency must match with the frequency modulation of exciting light to amplify the PA signal. However, temperature variation changes the thermal properties of PA cells, which induces shifts of the characteristic frequency. In order to obtain the drift distance of the characteristic frequency with the temperature increase, it was necessary to investigate the relationship between temperature and characteristics by controlling the characteristic frequency at the excited frequency via temperature compensation. We conducted a simulation of the original and optimized cavity based on the finite element method. The thermoviscous module of the cavities was sensitive to external and internal temperature variation. In our simulation, we enabled the temperature to range from 275 to 325 K with a step of 0.05 K by scanning the temperature of the air that was filling the opened cavities. Thus, we could acquire the characteristic frequency of the cavity at different temperature points. The relationship between temperature and characteristic frequency is illustrated in Figure 6, which shows that the characteristic frequency and temperature had a linear relationship in all cavities as the temperature increased from 275 to 325 K. The temperature sensitivities were 0.03, 0.042, 0.077, 0.09, and 0.092 kHz/K. It is obvious that the temperature sensitivities gradually increased with the starting frequency, and they were found to be in the same order of magnitude in a wide frequency range. Thus, temperature could induce the characteristic frequency shift no matter how a cavity is optimized or designed.

Fortunately, the influence caused by temperature can be compensated for due to the linear relationship between temperature and characteristic frequency, which paves the way for novel composite cavity design.

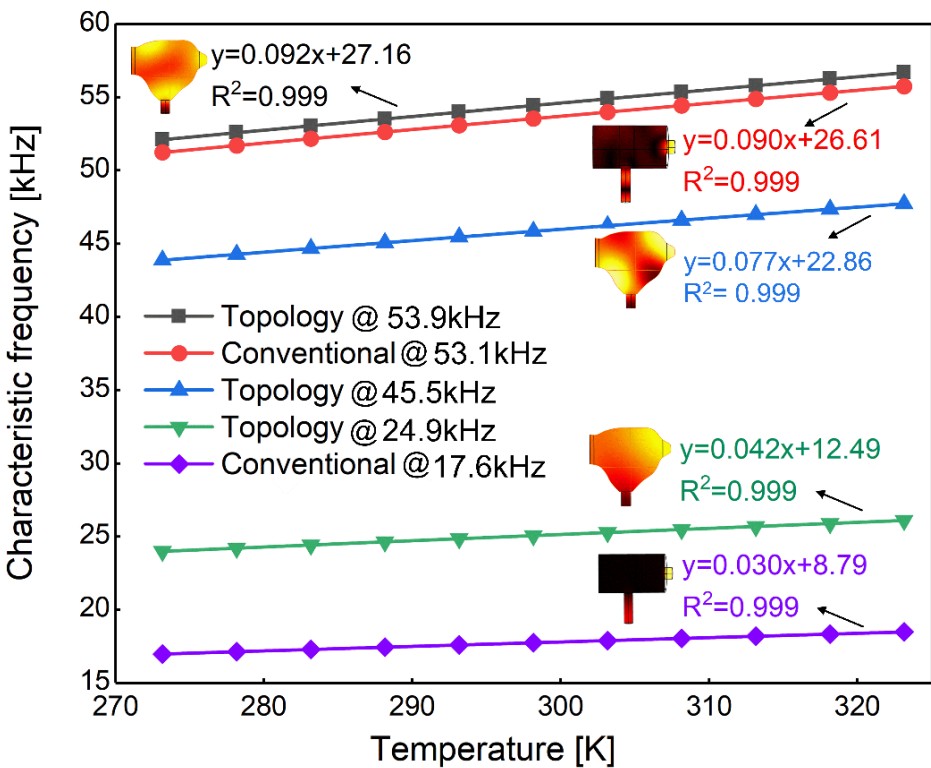

**Figure 6.** The influence of temperature on the characteristic frequency of two cells.

## 4. Conclusions

In summary, we introduced the theoretical topology optimization of a T-type opened resonator using a detailed finite element model analysis to noninvasively monitor glucose. Based on thermoviscous acoustics theory, we comparatively simulated the frequency response of optimized and conventional cells. Our comparison showed that the optimized cell had more resonance peaks over the frequency range of 0–70 kHz, and the PA signal gain was 2.7 and 1.3 times higher than the conventional cell around the frequency points of 25 and 52 kHz, respectively. The Q value was accordingly 1.9 and 4.9 times higher, respectively. Therefore, the detection sensitivity of PAS could be increased by our optimized cell. Our observations regarding temperature's influence on the characteristic frequencies, it shows that the characteristic frequencies of the two cells linearly increased with temperature. We believe that this topology optimization method will provide a new guide for developing PAS by optimizing the geometry of PA cells to obtain maximum signal gain regardless of the application scenario. In the future, we will explore and improve topology optimization technology in PA cells, and we will verify the performance of the optimized cell using further experiments.

**Author Contributions:** Z.T. and W.N. contributed to the idea. Z.T. contributed to the writing of the manuscript. Z.T., C.Y., W.N., P.P.S., S.C., J.H. and Z.L. contributed to the reviewing and editing of the manuscript. C.Y. supervised the project. All authors have read and agreed to the published version of the manuscript.

**Funding:** This work was supported by Natural Science Foundation of China (NSFC) under grants 62171487 and 62105373, Young and Middle Aged Talents Program of the State Ethnic Affairs Commission under grants MZR20004, and Key Technology R&D Program of Hubei province under grant 2020BBB097.

**Institutional Review Board Statement:** Not applicable.

**Informed Consent Statement:** Not applicable.

**Data Availability Statement:** All data generated or analyzed during this study are included in this published article.

**Conflicts of Interest:** The authors declare no conflict of interest.

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
