# Peer review of "Performance Enhancement of Opened Resonance Photoacoustic Cells Based on Three Dimensional Topology Optimization"

_photonics, doi:10.3390/photonics8090380_

Round 1
Reviewer 1 Report
All my questions have been answered very well. I reccomend to accept it for publication.
Reviewer 2 Report
Hi,
Only a minor comment related to the new added section: The authors shall remove "As we all know, characteristic ..." and replace it with " The characteristic...".
Best
This manuscript is a resubmission of an earlier submission. The following is a list of the peer review reports and author responses from that submission.
Round 1
Reviewer 1 Report
The authors use topology optimization models to enhance the performance of T-type open resonator PA cell. They constructed a thermo-acoustic model to simulate the frequency response over the range of 0-70KHz and the temperature field distribution. The simulations reveal an acoustic gain of the optimized cell of 2.7 times and 1.3 times higher 18 than the conventional cell.
This paper is probably publishable, but must be reviewed again after a major revision.
In the attached word file you can see some of the required modifications.

Author Response
We have carefully revised our paper (Manuscript ID: photonics-1306841) according to the comments of the reviewers and editors. Enclosed please find the reply to the reviewers’ comments. All the discussions have been properly added in the revised paper. We would like to thank the reviewers and editors for the kind help and valuable suggestions.

Reviewer 2 Report
The manuscript presents a topology optimization method to strengthen the whole performance on the conventional T-type opened PA cell. The simulation is quite clear and detailed. I think it has a very promising future in applications like noninvasively monitoring of glucose. I recommend to accept it with minor revisions.
1, if the dimention of the absorption cavity has any influence on the performance of the PAS cell?
2, the 'cardiac-type' cell shows a better performance according the simulation. However, is it possible to manufacture such a irregular shape? At least, it is not easy to polish.
3, the authors should pay attention on the captions of figures. The readers should have a clear understanding of each figure just via reading captions. For example, what's the meaing of 'PML' in Fig.1?
Author Response

(The authors gave the same response as above.)

Reviewer 3 Report
In this study the authors propose a new resonant cell for the PA methods. Although the MS has the merit for publication some points must be revised:
In Introduction there are many unreferenced statements from lines 36 to 44.
All work must be reviewed, and the introduction needs to be better referenced.
The main problem is why were the simulation range chosen from 0 - 75 kHz?
There are two operational problems with this range:
1 - The photoacoustic spectroscopy technique is generally measured at fixed frequencies and below 100 Hz. Unless the cells that are being analyzed will not be used specifically for photoacoustic (PA) spectroscopy but for Open Photoacoustic Cell (OPC).
The PA spectroscopy analyzes the PA signal as a function of the wavelength of the incident radiation. To this technique is interesting the resonant cell, since the PA signal from many samples, or at certain wavelengths, is very subtle.
The OPC analyzes the PA signal as a function of the modulation frequency of the incident radiation at a fixed wavelength, is used to obtain thermal parameters, such as thermal diffusivity. In this technique is not interesting the resonant cell, mainly due to calibration of OPC technique. The calibration is the function response of system (Microphone, PA cell and electronic amplification) in function of frequency modulation. The more linear the response, the better the measurement system, so, many resonances are not desired.
2 - Nowadays, the most used modulators for radiation sources when it comes to the OPC technique are electronics. Anyway, the maximum frequency reached is around 20 kHz.
Therefore, evaluating high frequencies may not be as relevant, making the cardiac-type cell with fewer resonances than the conventional one.
Another error is the use of the equation (8). This equation is an approximation for the thermally thin regime. Since this regime is dependent on frequency, the RG model equation without approximation should be considered.
Author Response

(The authors gave the same response as above.)

Round 2
Reviewer 1 Report
The authors must be aware that the SI abbreviation of Kilo is small "k" and not "K". Therefore, they must write kHz instead of KHz in the abstract and other positions in the manuscript!
Discussions of Figure 6 is still not clear to the reader. The author should try to answer the following questions while they discuss the figure findings: What do they want to investigate? Why do they want to investigate it? How they investigate it? Results of the investigation. Their comments and discussions on these result… As a reader, I should see the answer of these questions in your explanation.
The suggestions that I requested in the first version were examples. The paper still contains several English issues.
I still think that the paper is probably publishable, but several parts of the text are hard to understand. Therefore, a major revision is still required to improve the quality of the author’s papers. I will be glad to review it again!

Reviewer 3 Report
The authors improve the paper and presented adequate answers to the main questions raised. Therefore, I recommend the publication of the work.